# Determination of Parabens and Phenolic Compounds in Dairy Products through the Use of a Two-Step Continuous SPE System Including an Enhanced Matrix Removal Sorbent in Combination with UHPLC−MS/MS

**DOI:** 10.3390/foods12152909

**Published:** 2023-07-31

**Authors:** Laura Palacios Colón, Andrés J. Rascón, Evaristo Ballesteros

**Affiliations:** Department of Physical and Analytical Chemistry, E.P.S of Linares, University of Jaén, 23700 Linares, Jaén, Spain; lpcolon@ujaen.es (L.P.C.); arascon@ujaen.es (A.J.R.)

**Keywords:** phenolic compounds, parabens, dairy products, continuous solid-phase extraction, EMR-lipid sorbent, UHPLC−MS/MS

## Abstract

Dairy products can be contaminated by parabens and phenolic compounds from a vast variety of sources, such as packaging and manufacturing processes, or livestock through feed and environmental water. A two-step continuous solid-phase extraction (SPE) and purification methodology was developed here for the determination of both types of compounds. In the first step, a sample extract is passed in sequence through an EMR-lipid sorbent and an Oasis PRiME HBL sorbent to remove fat and preconcentrate the analytes for subsequent detection and quantification by UHPLC−MS/MS. This method enabled the determination of 28 parabens and phenolic contaminant with excellent recovery (91–105%) thanks to the SPE sorbent combination used. The proposed method was validated through the determination of the target compounds, and was found to provide low detection limits (1–20 ng/kg) with only slight matrix effects (0–10%). It was used to analyse 32 different samples of dairy products with different packaging materials. Bisphenol A and bisphenol Z were the two phenolic compounds quantified in the largest number of samples, at concentrations over the range of 24–580 ng/kg, which did not exceed the limit set by European regulations. On the other hand, ethylparaben was the paraben found at the highest levels (33–470 ng/kg).

## 1. Introduction

Dairy products are highly complex foods rich in micro- and macro-nutrients that are massively consumed in Spain. Based on the latest official food consumption report, milk is consumed at a rate of 71.27 L/person each year in the country [1]. Milk and its derivatives are exposed to contamination from the environment, livestock feed, and packaging materials. Environmental pollution is a source of major concern as it jeopardizes the sustainability of the planet. Industrialization, the rapid growth of the human population, and anthropogenic activities generate contamination that pollutes the ecosystem [2].

Parabens, phenolic compounds, and various other pollutants can reach milk through cattle (cows, goats, and sheep) being directly exposed to environmental or feeding contaminants [3]. The long-term consumption of foods of an animal origin containing residual contaminants can have toxic effects on the human body through bioaccumulation [4]. Some compound families have been deemed endocrine disruptor chemicals (EDCs) because they can interfere with biosynthetic processes and hormonal action. Most EDCs’ phenolic compounds and the bisphenols included come from anthropogenic sources [5,6].

Parabens are alkyl esters derived from *p*-hydroxybenzoic acid, used by the cosmetic, pharmaceutical, food, and beverage industries for their antimicrobial and preservative properties [6,7,8]; however, phenolic compounds are highly toxic and persistent in the environment, so they pose some risks for living organisms. Phenolics are essential in the production of pesticides, pharmaceuticals, packaging materials, household preservatives, and as intermediates in industry [9]. For example, triclosan is usually added to body care products such as liquid and solid soap, toothpaste, and mouthwash. Its antimicrobial properties also make it useful for food contact materials, where it can account for as much as 1% (*w*/*w*) of their contents, so it has been banned for use in medical devices, for example [10]. Bisphenol A (BPA) is a synthetic phenol used to obtain polycarbonate plastics and epoxy resins for the production of a number of consumer products [8,11,12]. BPA is suspected to cause health issues such as cancer and reproductive disorders, and also to be an obesity promoter [9,13]. On these grounds, the European Food Safety Authority (EFSA) recently downgraded its tolerable daily intake (TDI) 100,000 times (from 4000 ng/kg to 0.04 ng/kg) [14]. In contrast, the European Commission set the maximum specific migration limit (SML) for BPA at 0.05 mg/kg in 2018 to update the previous 2011 level [15]. BPA has been already banned in food contact materials intended for children, which has forced manufacturers to use alternative compounds such as bisphenol B, F, or S. As a result, these BPA analogues, which are similarly toxic to the parent compound or even more so, are now ubiquitous in environmental and biological samples, and the exposure of humans (through the consumption of dairy products, for example), other living organisms, and the environment has increased with time [16]. Because of its similarity in structure, toxicity, and metabolism, BPF has been deemed an unsafe alternative to BPA [17]. So far, the European Union has set the SML for only one BPA analogue, bisphenol S, at 0.05 mg/kg [18].

The US Environmental Protection Agency (EPA) know about the potential threat of phenol pollutants and has included the following to its Priority Pollutant List: 4-chloro-3-methylphenol, 2,4,6-trichlorophenol, 2,4-dichlorophenol, pentachlorophenol, 2,4-dinitrophenol, 2-methyl-4,6-dinitrophenol, 4-nitrophenol, phenol, 2-nitrophenol, and 2,4-dimethylphenol [19]. In addition, it has classified pentachlorophenol as a group B2, probable human carcinogen [13]. On the other hand, European Commission (EC) Regulation No. 10/2011 has set the SML for phenolics such as 4-*tert*-butylphenol, phenol and 2,6-dimethylphenol at 0.05 mg/kg for specific migration from food contact materials [18]. Nonylphenol and 4-octylphenol can disrupt reproductive and immune functions, and have adverse health effects as a result [12]. The former has been included as a priority contaminant on EPA’s Candidate Contaminant List (CCL4) [20].

Parabens and phenolic compounds in dairy products are most often determined by gas or liquid chromatography. Gas chromatography (GC) is an effective choice for determining EDCs by virtue of its good separation efficiency and high throughput. For example, González et al. [21] and Palacios et al. [22] determined ECDs in dairy products by using GC coupled to a mass spectrometer (GC−MS), whereas Zhang et al. determined the phenolics in milk by GC with flame ionization detection (GC−FID) [23]. These compounds are also frequently determined by high-performance liquid chromatography (HPLC). The main advantage of HPLC over GC for phenolics is that the analytes need not be derivatized. Santonicola et al. [17], Tuzimski and Szubartowski [16], and Xiong et al. [24] used HPLC with fluorescence detection (HPLC−FLD) to determine the bisphenols in dairy products. However, as can be seen in Table 1, liquid chromatography coupled to tandem mass spectrometry (HPLC−MS/MS) is the most frequently employed analytical technique to monitor phenol compounds, bisphenols, and parabens, a result of its affording detection at trace levels in dairy products [7,25,26,27,28].

Sample pre-treatment is a fundamental step in the determination of the previous types of compounds, especially when analytes are present at low concentrations in complicated matrices such as dairy products. Solid-phase extraction (SPE) provides a highly efficient, simple, and economical method to enrich samples with the target analytes for their subsequent determination [29]. As can be seen in Table 1, SPE has been used with various sorbent materials to extract parabens and phenolic pollutants from milk and dairy products [17,28,30]. In addition, a new generation of easily used, robust sample preparation sorbents with virtually no matrix effects, including Oasis PRiME HLB, has recently been developed [31]. This is a reversed-phase SPE sorbent that is packed in a column where the sample is directly passed through after extraction, saving time, and it is also more environmentally friendly than the conventional procedure and can remove the majority of matrix interferents such as proteins, salts, and phospholipids [4]. Enhanced matrix removal-lipid (EMR-lipid), which was introduced by Agilent Technology in 2015, is another recently developed sorbent material especially designed for lipid removal. In fact, EMR-lipid selectively clear major lipids from samples avoiding undesired analyte retention [16,27,32,33].

As can be seen in Table 1, there are other techniques in the literature for the treatment of milk and dairy product samples, such as QuEChERS [24,25,27], matrix solid-phase dispersion [7], dispersive magnetic micro-solid phase extraction [34], electrochemical assistance solid-phase microextraction [23], and ultrasound-assisted liquid−liquid microextraction [35].

In previous work, our group determined EDCs using GC−MS [22], which required derivatizing the analytes to boost analytical signals and avoid false positives. In this work, we determined seven parabens and twenty-one phenolic compounds classified as EDCs in dairy products by using ultra-high-performance liquid chromatography−tandem mass spectrometry (UHPLC−MS/MS) to expedite sample treatment and avoid the need for organic solvents. This technique is one of the best choices for quantitative analysis thanks to its high sensitivity and reproducibility, confirmed by the two most abundant product ions. EMR-lipid was used in combination with Oasis PRiME HLB in a closed semi-automated SPE system to remove fat and protein from the sample matrix and increase the recovery of the target analytes as a result. The main goal was to simplify and unify the extraction procedure for the target compounds while improving the efficiency of the process and reducing the matrix effects. The resulting SPE−UHPLC−MS/MS method will be evaluated in terms of sensitivity, selectivity, precision, and recovery, after which it will be used to determine the target analytes in dairy products packed in different kinds of food contact materials (namely, high density polyethylene (HDPE), polystyrene (PS), polyethylene terephthalate (PET), multilayer packaging, and glass) to check whether any of them were responsible for the presence of the analytes in any sample.

**Table 1 foods-12-02909-t001:** Selected references for the determination of parabens and phenolic compounds in dairy products.

Analytes ^a^	Samples	Sample Treatment ^b^	Technique ^c^	Analytical Figures of Merit ^d^	Concentrations Found in Real Samples	References
BPA, parabens, and other plasticisers	Milk	MSPD	HPLC–MS/MS	LOQ 0.91–4.4 ng/kg RSD < 14%	BPA 1.8–59 ng/kg	[7]
BPF, BPE, BPB, BPA, and diglycidyl ether	Breast milk	SPE, QuEChERS	HPLC–FLD	LOD 57,000–78,000 ng/L RSD 1.0–14% R 57–88%	n.d.	[16]
BPA and BPF	Milk	SPE	HPLC–FLD	LOD 30 ng/L RSD 1.0–14.8% R 98–107%	BPF 1020–2690 ng/L	[17]
BPA, BPB, BPF, BPZ, BPS, and other bisphenol analogues	Yogurt	QuEChERS	CG–MS	LOQ 170 ng/kg	n.d.	[21]
Phenol, alkyphenols, chlorophenols, and bisphenols	Dairy products	SPE	CG/MS	LOD 6–63 ng /L RSD 2.4–11% R 85–108%	BPA 30–1400 ng/kg BPZ 96–1100 ng/kg BPF 270–950 ng/kg Nonylphenol 56–390 ng/kg 4-*tert*-Butylphenol 310–2100 ng/kg 3.4-Dimethylphenol 130–2300 ng/kg 4-Pentylphenol 190–990 ng/kg 2-Phenylphenol 320 ng/kg	[22]
4-Chlorophenol, 4-*tert*-butylphenol, 4-*tert*-octylphenol, and other phenols	Milk	EA–SPME	GC–FID	LOD 1–30 ng/L RSD 3.8–12.4% R 88–119%	4-*tert*-Butylphenol 850 ng/L 4-Chlorophenol n.d. 4-*tert*-Octylphenol n.d.	[23]
BPA, BPB, and other bisphenol analogues	Milk	QuEChERS	HPLC–FLD	LOD 1000–3100 ng/L RSD 9.1–16% R 86–99%	BPA 1374 ng/kg	[24]
BPA, BPF, BPS, and parabens	Breast milk	QuEChERS	HPLC–MS/MS	LOQ 100–250 ng/L RSD 5–16% R 83–115%	BPA 130–1 620 ng/L BPF 130–320 ng/L BPS 370 ng/L Parabens 130–4050 ng/L	[25]
BPA, nonylphenol, and 4-*ter*-octylphenol	Milk	MSPD	HPLC–MS/MS	LOD 50–100 ng/kg RSD 1.4–7.8% R 84–103%	BPA 490 ng/kg Nonylphenol 100 ng/kg 4-*ter*-Octylphenol 4240–17,600 ng/kg	[26]
BPA, BPB, BPF, BPS, and other bisphenol analogues	Breast milk and sweetened condensed milk	QuEChERS	HPLC–DAD/ HPLC–MS/MS	LOQ 100–250 ng/L RSD 8–17% R 35–102%	BPA 230–690 ng/L BPS 290–680 ng/L BPF 220–290 ng/L	[27]
Parabens, TCS, BPA, nonylphenol, 4-*tert*-octylphenol, and other phenols	Breast milk	SPE	CG/MS	LOD 1–9 ng /L RSD 4.4–7.0% R 86–104%	Phenols 550–5600 ng/L BPA 1400–2900 ng/L Parabens 15–8100 ng/L	[30]
BPA	Milk	DMSPE	HPLC–UV	LOD 3 050 ng/L RSD 9.1–16% R 86–99%	n.d.^e^	[34]
BPA, BPF, BPAF, and 4-chlorophenol	Milk	UA–DLLME	HPLC–UV	LOD 250–1000 ng/L RSD 0.67–13.7% R 83–119%	n.d.	[35]
BPA and BPS	Milk	LLE	PSI–MS	LOD 1500–4800 ng /L RSD 5.9–18.9% R 74–112%	BPA 77,600–150,800 ng/L BPS 60,000 ng/L	[36]
Nonylphenol, 4-*tert*-octylphenol and BPA	Dairy products	UA–DLLME	HPLC–FLD	LOD 10–40 ng/kg RSD 2.8–8.8% R 83–112%	BPA 800–128,700 ng/kg Nonylphenol 3500–36,700 ng/kg 4-*tert*-Octylphenol 300–29,500 ng/kg	[37]
BPA, BPB, BPF, BPZ, 4-*tert*-octylphenol, and nonylphenol	Milk	QuEChERS	UHPLC–MS/MS HPLC–FLD	LOD 200–1500 ng/L	BPA 500–5600 ng/L BPF 500–8700 ng/L	[38]
BPA, BPS, BPAF, BPF and parabens	Breast milk	QuEChERS	HPLC–MS/MS	LOD 10–200 ng/L RSD 1.0–7.9% R 77–100%	BPA 990–1910 ng/L BPS 60–200 ng/L BPAF 60–70 ng/L BPF n.d. Parabens 30–1520 ng/L	[39]
Parabens, phenol, alkylphenols, chlorophenols and bisphenols	Dairy products	Continuous SPE	HPLC–MS/MS	LOD 1–20 ng/kg RSD 2.0–11.4% R 91–105%	BPA 33–580 ng/kg BPZ 24–57 ng/kg Nonylphenol 34–210 ng/kg 4-*tert*-Butylphenol 21–220 ng/kg 4-Chlorophenol 50–160 ng/kg Pentachlorophenol 35–76 ng/kg Parabens 9–470 ng/kg	This work

^a^ BPA, bisphenol A; BPAF, bisphenol AF; BPB, bisphenol B; BPE, bisphenol E; BPF, bisphenol F; BPS, bisphenol S; BPZ, bisphenol Z. ^b^ DMSPE, dispersive magnetic solid-phase extraction; EA–SPME, electrochemical assistance solid-phase microextraction; MSPD, matrix solid-phase dispersion; QuEChERS, quick, easy, cheap, effective, rugged, and safe; SPE, solid-phase extraction; SPME, solid-phase micro-extraction; UA–DLLME, ultrasound–assisted dispersive liquid–liquid microextraction; VASUSME, vortex-assisted supramolecular solvent microextraction. ^c^ HPLC–DAD, high-performance liquid chromatography–photodiode array detection; HPLC–FLD, high-performance liquid chromatography with fluorescence detection; HPLC–MS/MS, high-performance liquid chromatography coupled to triple-quadrupole mass spectrometry; HPLC–UV, high-performance liquid chromatography with ultraviolet detection; GC–FID, gas chromatography with flame–ionization detection; GC–MS, gas chromatography coupled to single-quadrupole mass spectrometry; MSPD, matrix solid-phase dispersion; PSI-MS, paper spray ionization−mass spectrometry. ^d^ LOD, limit of detection; LOQ, limit of quantification; R, recovery; RSD, relative standard deviation. ^e^ n.d., not detected.

## 2. Materials and Methods

### 2.1. Reagents and Chemicals

HPLC-grade water, methanol (MeOH), acetone, acetonitrile (ACN), and *n*-hexane were supplied by Sigma-Aldrich (St. Louis, MO, USA). Formic acid (FA, 98% purity) was purchased from Fluka (St. Louis, MO, USA), and MgSO_4_ and NaCl were both obtained in the highest available purity from PanReac AppliChem (Barcelona, Spain). 

Analytical standards of parabens (metylparaben, ethylparaben, propylparaben, isopropylparaben, butylparaben, isobutylparaben, and benzylparaben), phenolic compounds (3,4-dimethylphenol, 2,5-dimethylphenol, 4-*tert*-butylphenol, 2-*tert*-butyl-4-methylphenol, 4-pentylphenol, 2-phenylphenol, 4-phenylphenol, 4-hexylphenol, 4-heptylphenol, 4-chlorophenol, 4-chloro-3-methylphenol, pentachlorophenol, triclosan, 4-*tert*-octylphenol, nonylphenol, BPA, bisphenol F, bisphenol B, bisphenol S, and bisphenol Z), and triphenylphosphate (internal standard (IS)) were supplied by Sigma-Aldrich (St. Louis, MO, USA) in the highest available purity.

Centrifuge tubes were obtained from J.D. Catalan S.L. (Madrid, Spain). Regarding the sorbents, Oasis HLB and Oasis PRiME HLB were acquired from Waters Corporation (Milford, MA, USA), and Captiva EMR-Lipid was acquired from Agilent Technologies (Santa Clara, CA, USA).

All of the analytical standards were made using 5 g/L in MeOH. The standard solutions were stored in glass-stoppered bottles at 4 °C in the dark. Mixed standards with all of the analytes were prepared at 1 mg/L in MeOH on a daily basis and diluted when needed. The SPE eluent was 30:70 (*v*/*v*) ACN:H_2_O containing of 400 μg/L of IS and was prepared daily.

### 2.2. Equipment 

For separation, a Dionex Ultimate 3000 UHPLC chromatograph (Thermo Fisher Scientific, Waltham, MA, USA) was used equipped with Zorbax Rapid Resolution High Definition (RRHD) Eclipse Plus C18 column (2.1 mm × 100 mm, 1.8 μm particle size) from Agilent Technologies (Santa Clara, MA, USA). The mobile phases were ultrapure water (Solvent A) and acetonitrile (Solvent B), both containing 0.1% of formic acid; the flow was set at 0.4 mL/min. The column oven was set at 40 °C and the injection volume was set to 10 µL. The elution programme was as follows: 10% B (0–2 min), 10–30% B (2–18 min), 30–40% B (18–24 min), 40–70% B (24–32 min), and 70–10% B (32–35 min). The UHPLC equipment was coupled to a TSQ Endura triple quadrupole (QqQ) from Thermo Scientific (USA) equipped with a heated electrospray ionization probe (HESI) set in the positive ion mode. The spray voltage was 3500 V; the sheath and aux gas settings were 45 and 13 arbitrary units, respectively; the ion transfer tube and vaporizer temperature were 400 and 350 °C, respectively; and the collision gas (CID) pressure was 2 kPa. The individual analytes were infused and characterized in the mass spectrometer triple quadrupole equipment in multiple reaction monitoring mode (MRM) (Table 2).

Other equipment used included the continuous solid-phase extraction system, composed of a Gilson peristaltic pump (Villiers-le-Bel, France), three injection valves of model 5041 from Rheodyne (Cotati, CA, USA), poly (vinyl chloride) tubing, PTFE columns that were custom-packed with the sorbent materials (Oasis PRiME HLB and EMR-lipid), and a Centrofriger BL-II centrifuge from JP Selecta (Barcelona, Spain). The Oasis PRiME HLB sorbent column was conditioned by passing 1 mL of methanol, 1 mL of acetonitrile, and 2 mL of Milli-Q water in this sequence. The EMR-lipid sorbent column was conditioned by passing 2 mL of Milli-Q water.

### 2.3. Sampling

Thirty-one samples of dairy products including fifteen samples of milk (skimmed, semi-skimmed, and whole milk from cow; fresh cow milk; semi-skimmed goat milk; and semi-skimmed sheep milk) and sixteen dairy products samples (yogurts, custards, milkshakes, cheeses, creams, butters, and margarines) were analysed. The samples represent a wide range of the most consumed dairy products in Spain and were obtained from different commercial brands in containers made of various materials such as polyethylene terephthalate (PET), high-density polyethylene (HDPE), polystyrene (PS), multilayer material (a mixture of cardboard, polyethylene (PE) and aluminium), and glass. All of the samples were subjected to pasteurization, UHT, or sterilization for preservation. In addition, all were bought from Spanish supermarkets, kept under the conditions recommended on their packaging, and stored in the dark at 4 °C until analysis.

### 2.4. Sample Treatment

The sample pre-treatment of high-throughput screening methods usually involves retaining and extracting the target compounds as far as possible while removing impurities such as protein and fat to the greatest extent [4]. In this work, we developed an extraction procedure involving liquid−liquid extraction followed by clean-up and preconcentration of the extract with two solid-phase sorbents (EMR-lipid and Oasis PRiME HLB) in a serially connected semiautomated SPE system. The analytes were determined by UHPLC−MS/MS.

First, the samples were homogenized in 50 mL propylene centrifuge tubes. Then, 1 g of each sample (milk, milkshakes, custards, yogurts, cream, cheese, margarines, and butters) was transferred to another tube and supplied with 8 mL of ACN containing 0.5% FA (*v*/*v*), as previously described by our group [22]. The tubes were then centrifuged at 5000 rpm (2150 g) at 4 °C for 10 min, and the resulting supernatant was collected and supplied with 1.6 g MgSO_4_ and 0.4 g NaCl, vortexed for 2 min, and centrifuged at 5000 rpm (2150 g) at 4 °C for 2 min.

The final extract was diluted to 50 mL with ultrapure water and its pH adjusted to 4 for passing through the closed, semi-automated SPE system packed with two serially columns of sorbent material. The first column contained 50 mg of EMR-lipid and the second 70 mg of Oasis PRiME HLB. The extract was introduced into the system at a flow rate of 5.0 mL/min to clean it up and allow the analytes to be retained on the sorbents, respectively, while the matrix and interferents were sent to waste (Figure 1). The Oasis PRiME HLB sorbent was then dried by an air stream circulating at a flow of 5 mL/min in both directions for 3 min and the analytes were then eluted with 600 µL of a 30:70 (*v*/*v*) ACN:H_2_O mixture containing a 400 µg/L concentration of IS. The vials were hermetically sealed and a volume of 10 µL was injected into the UHPLC−MS/MS system for analysis. Appendix A details the operation of the continuous system for the SPE.

## 3. Results and Discussion

### 3.1. Sample Treatment Optimization

The good performance of a method for determining analytes at trace levels is largely dependent on how the samples are pre-treated. The matrices of dairy products are very complex due to their high fat (0.3–82%) and protein content (0.5–24%), thus requiring very efficient treatment procedures [40].

We initially used a liquid−liquid extraction (LLE) procedure previously optimized by our group [22]. For this purpose, an amount of 1 g of sample was spiked with a 500 ng/kg concentration of each analyte and subjected to LLE with 8 mL of acetonitrile containing 0.5% FA (*v*/*v*). SPE, which is one of the most commonly used techniques for the clean-up of extracts and for the preconcentration of analytes [17,41]. Because the analytes were highly soluble in fats, their presence in the extracts could detract from recoveries as the analytes might be co-discharged to waste with the fat fraction. The efficiency with which fat and other interferents were removed from the samples was assessed by processing the supernatant from the first step in the following ways: (1) direct analysis of the extract, (2) passage through an SPE system with the EMR-lipid sorbent, and (3) passage through an SPE system using a combination of EMR-Lipid and Oasis PRiME HBL. The efficiency of the sorbent was evaluated in terms of recovery. As can be seen in Figure 2, recoveries without EMR-lipid ranged from 20 to 120% for milk and 60 to 200% for butter. Passing the sample through the column containing EMR-lipid improved the recoveries somewhat (62–117% from milk and 42–163% from butter), although inadequately, and required introducing a third cleaning/preconcentration step involving passing the extract through a second column packed with Oasis PRiME HLB. Using the two sorbents substantially improved recoveries, which ranged from 90 to 105% for both milk and butter. Similar results were obtained with the other types of dairy products. Therefore, all further tests were conducted using a column packed with EMR-Lipid and then one packed with Oasis PRiME HBL.

The optimum amount of EMR-Lipid sorbent to be used to remove fat from the LLE extract was established by passing it through columns containing variable amounts of sorbent (20–80 mg). Amounts exceeding 40 mg were found to provide near-quantitative analyte recoveries. This led us to choose 50 mg of EMR-Lipid for further testing. The optimum amount of the second sorbent, Oasis PRiME HBL, was established similarly. A number of calibration graphs were run with different sorbent columns packed with 20–120 mg of sorbent to pass 100 mL of standard aqueous solutions with variable concentrations of each target compound over the range of 50–500 ng/L. The peak areas of the analytes increased with the increased amount of sorbent up to 60 mg, but decreased above 80 mg owing to the need for a higher volume of eluent to ensure complete elution of the analytes. Based on these results, columns packed with 70 mg of Oasis PRiME HBL were selected for further tests.

On the other hand, several solvents or a mixture of solvents (MeOH, EtOH, ACN, acetone, MeOH:H_2_O, and ACN:H_2_O) were tested as eluents of the analytes adsorbed in the continuous SPE system. For this purpose, 1 g of milk, milkshake, custard, heavy cream, yogurt, cheese, butter, or margarine was spiked with a 500 ng/kg concentration of each analyte and subjected to the treatment described in Section 2.4. The best results were obtained using the ACN:H_2_O mixture, at a ratio of 30:70 (*v*/*v*), ensuring near-quantitative elution against only 10–40% with the other choices, also the effect of the volume used on the elution was examined in the range of 50–1000 µL. It was observed that for volumes above 550 µL, the elution of the analytes retained on the Oasis PRiME HBL column was quantitative. An eluent volume of 600 µL was thus selected for further testing. It was also studied whether the sample flow rate affected the retention of analytes in the sorbent column. It could be seen that the sample flow rate that passed through the column had no effect on the sorption efficiency in the studied range (0.5–6.0 mL/min). Thus, 5.0 mL/min was selected as the sample and air flow rate. Finally, the sample breakdown volume was tested by using a determined amount of each compound (20 ng) in different volumes from 10 to 500 mL. Volumes up to 200 mL resulted in near-quantitative sorption with both Oasis PRiME HBL and EMR-Lipid.

The sample pH intensely influenced the efficiency with which the parabens and phenolic compounds were retained by the SPE sorbent. Tests were conducted over the pH range of 1–10 by passing 50 mL of aqueous solutions containing 500 ng/L of each target compound adjusted with diluted HCl or NaOH. It was found that the extraction yields of the analytes were the highest in the range between pH 3 and 5, and then diminished at higher values, so pH 4 was selected for further testing.

### 3.2. UHPLC−MS/MS Analysis

All of the analytes were infused and characterized by optimizing the multiple reaction monitoring (MRM) transitions for each one. As can be seen in Table 2, the precursor ion of each analyte was detected and two product transitions were optimized. The primary transition involved the most abundant product ion, which was used as the quantifying ion (Q), and a secondary ion that was used as the confirming ion (q). The procedure was applied with the software XCalibur 3.0.63 from Thermo Fisher Scientific (San José, CA, USA) and the data were processed with TraceFinder 3.2.5.12.0, also from Thermo Fisher Scientific. Table 2 shows the values of the MRM transitions, optimized collision energy, and lens voltage. The final detection conditions are described in Section 2.2.

### 3.3. Validation of the Method

The proposed methodology was validated by evaluating its performance with the most representative samples (milk, custard, yogurt, milkshakes, cheese, cream, butter, and margarine). The performance was assessed in terms of selectivity, linearity, matrix effects, precision, and recovery. These quality-related properties were determined by running calibration curves with a blank milk sample that was spiked with the analytes at variable concentrations and analysed as described in Section 2.4. Table 2 shows the most salient results. The method provided a linear response over the concentration range of 3.3–20,000 ng/kg and the correlation coefficients (*r*^2^) exceeded 0.994 for all of the analytes. Limits of detection (LODs) and quantification (LOQs) were calculated in terms of the signal-to-noise ratio (S/N = 3 for LOD and S/N = 10 for LOQ). As can be seen from Table 2, LODs ranged from 1.0 to 20 ng/kg and LOQs coincided with the lower limit of each linear range. The limits of quantification of the proposed method varied between 3.3 and 65 ng/kg. These values were lower than the EU regulatory limits of the legislated analytes, as previously indicated [18]. Similarly, the LOQs of several of the methods included in Table 1 were lower than the EU regulatory limits [7,25,27].

The precision of the method was assessed in terms of the relative standard deviation (RSD), both intraday and interday, by spiking representative uncontaminated samples (*n* = 12) with the analytes at three different concentration levels (100, 500, and 2000 ng/kg). As can be seen from Table 3 and Appendix A intraday precision was 2.03–8.91% and interday precision was 5.21–11.44%, so both were good and on a par with those for alternative methods [34,38].

Matrix effects (*ME*) can arise during chamber ionization (HESI) as a result of co-extractives interfering with the electrospray ionization of the analytes and with an adverse impact on precision [32]. *ME* is a measure of the deviation of the slope of the matrix calibration curve from that of the standard curve [39]:$$ME=\left[\left(\frac{slope\;of\;matrix-matched\;curve}{slope\;of\;in-solvent\;curve}\right)-1\right]\cdot 100$$

The effects were classified as negligible [(0%)–(±10%)], mild [(±10%)–(±20%)], medium [(±20%)–(±50%)], or strong (>±50%). As shown in Appendix A and Figure 3, ME ranged from 1 to 10%, so the matrix effects were negligible. These results are better than the previously reported values (1–20%), which can be ascribed to our using the EMR-Lipid sorbent [22]. Therefore, using EMR-Lipid in combination with Oasis PRiME HLB efficiently reduced the matrix effect, as can be seen in Appendix A.

The accuracy of the method was evaluated in terms of recovery, using samples spiked with a mixture of target compounds at three different concentration levels (100, 500, and 2000 ng/kg) that were processes as described in Section 2.4. The analyte recoveries thus obtained from the triplicate determinations (*n* = 3) are shown in Table 4 and Appendix A. As can be seen, they ranged from 91 to 105%. These recoveries were also better than those obtained in previous work without the EMR-Lipid sorbent (85–108%) [22]. Dualde et al. [25], Tuzimski and Szubartowski [27], and De Almeida Soares et al. [42] used alternative extraction and determination techniques for bisphenols and parabens in dairy products and obtained recoveries of 83–115%, 15–103%, and 74–112%, respectively.

### 3.4. Application to Real Samples

Once the proposed the method was checked whether it performed well regarding sensitivity, recovery, precision, and matrix effects, it was applied to real samples. Thus, it was used to determine seven parabens and twenty-one phenolic compounds in fifteen samples of milk (skimmed, semi-skimmed and whole milk from cow; fresh cow milk; semi-skimmed goat milk; and semi-skimmed sheep milk) and sixteen samples of dairy products (milkshakes, yogurts, cheeses, butters, margarines, and creams).

All samples were analysed in triplicate and a blank was introduced after each three to discard potential contamination from other sources. Samples were chosen in terms of representativeness on the market and spanned various packaging materials including high-density polyethylene (HDPE), multilayer-cardboard (MC), polyethylene terephthalate (PET), polystyrene (PS), and glass.

As can be seen from Table 5 and Table 6, the most frequently detected contaminants were BPA (68% of samples) and BPZ (39%). BPA was found at concentrations over the range 33–580 ng/kg, especially in whole cow milk and milkshakes held in MC, PET, or HDPE. In contrast, it was completely absent from yogurt samples held in glass. De Almeida Soares et al. previously detected BPA at concentrations of 77,600–150,800 ng/L in milk [38], and Lv et al. found it at levels from 800 to 128,700 g/kg in dairy products [37]. Azzouz et al. detected BPA at concentrations from 1400 to 2900 ng/L in human milk, thus demonstrating that contamination from food and the environment can reach human bodies [30].

BPZ was the second most detected contaminant at 24 to 57 ng/kg. However, as displayed in Table 5 and Table 6, it was only detected in milk samples, with its peak concentrations in the samples held in multilayer cardboard (MC) and its lowest levels in those held in HDPE. No BPZ analogues, such as BPF, BPS, or BPB were detected. In contrast, De Almeida Soares et al. [42] and Di Marco Pisciottato et al. [38] found BPS at concentrations of 60,000 ng/L and BPF at levels of 500–8700 ng/L, respectively, in milk.

Nonylphenol is one other frequently monitored contaminant. In 2022, it was included into the candidate contaminants for the regulatory determination list [20]. This phenolic compound was found in 26% of our samples, at concentrations over the range of 34–210 ng/kg in milk packed in HDPE, PET, and MC, and in both butter and margarine packed in PS. As displayed in Table 1, Lv et al. detected nonylphenol at concentrations of 3500–36,700 ng/kg in dairy products [37] and Shao et al. found it at 100 ng/kg in milk samples [26].

4-*tert*-Butylphenol was spotted in 19% of the milk samples (concentrations of 21–220 ng/kg), and 4-chlorophenol (50–160 ng/kg) and pentachlorophenol (35–76 ng/kg) both in 16%. As shown in Table 5, these contaminants were present in the majority of samples of milk packed in MC. Zhang et al. previously detected 4-*tert*-butylphenol at 850 ng/kg in milk [23].

Parabens are employed as food preservatives for their anti-microbial and anti-fungal properties. As can be seen in Table 3, only three of the parabens studied were detected in the samples. The following were only found in milk, regardless of the packaging material: ethylparaben (33–470 ng/kg), propylparaben (9–33 ng/kg), and benzylparaben (24–130 ng/kg). Dualde et al. [25] and Azzouz et al. [30] detected parabens at concentrations of 130–4050 ng/L and 15–8100 ng/L, respectively, in breast milk.

## 4. Conclusions

A new method for the simultaneous determination of 21 phenol compounds and 7 parabens in various types of dairy products was developed. The method combines conventional LLE extraction with SPE in a closed semi-automated system furnished with two different SPE sorbents (ERM-lipid and Oasis PRiME HLB) to avoid unwanted matrix effects, and increase analyte recoveries and expedite sample treatment. This substantially reduced the consumption of samples and solvents, as well as the analysis time, relative to the existing methods for the same purpose. Thus, the proposed method speeds up the sample treatment by reducing the time of this step to 15 min compared with other methods that need more than 40 min [4,24,25]. The optimized UHPLC−MS/MS method allowed the target analytes to be detected and quantified with good precision (RSD 2–11.4%), low matrix effects (0–10%), and high recoveries (91–105%). These results are quite good if one considers the high complexity of the sample matrices. A total of 31 samples (15 of milk and 16 of dairy products) were analysed. The most detected phenolic compounds were BPA (33–580 ng/kg) and BPZ (24–57 ng/kg). In the case of BPA, these concentration levels did not exceed the legal limit set by the European Union. Nonylphenol was found in 26% of the samples, at concentrations up to 210 ng/kg. The most detected paraben was ethylparaben, which was found at levels up to 470 ng/kg. Glass packaging was found to hold the least contaminated samples and plastics the most contaminated ones (particularly milk).

## Figures and Tables

**Figure 1 foods-12-02909-f001:**
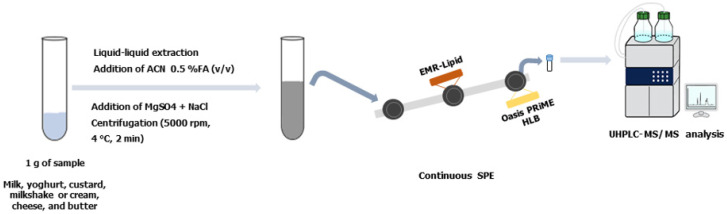
Procedure for determining parabens and phenolic compounds in dairy products using UHPLC−MS/MS.

**Figure 2 foods-12-02909-f002:**
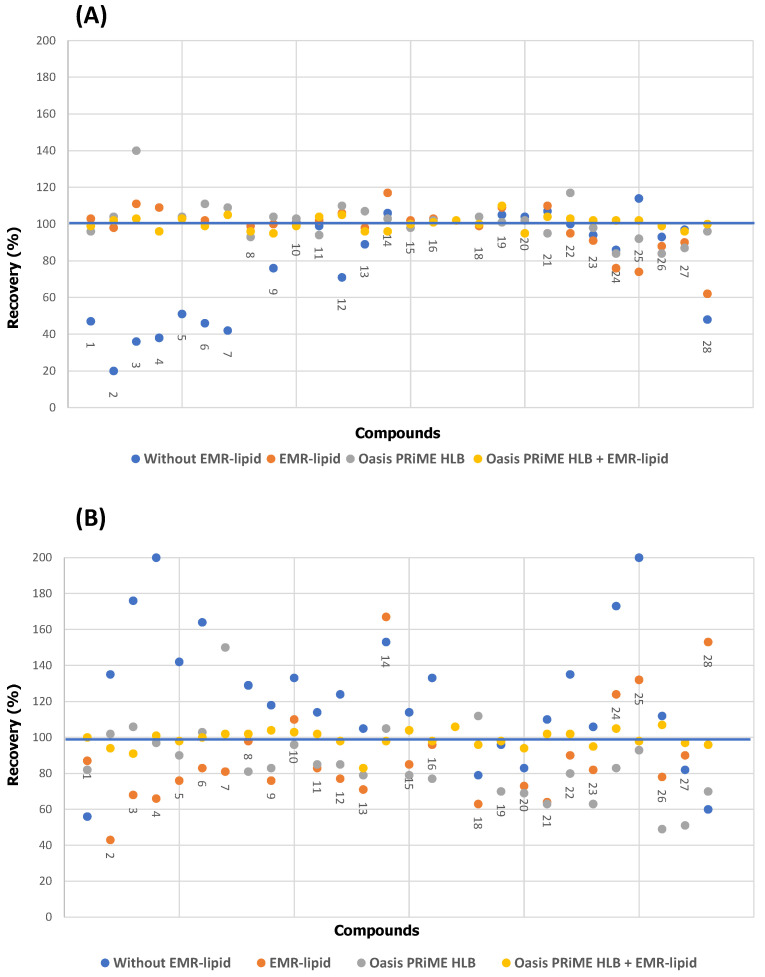
Recovery of parabens and phenolic compounds from milk (**A**) and butter (**B**) using different sorbents: no EMR-lipid, EMR-lipid, Oasis PRiME HLB, and EMR-lipid + Oasis PRiME HLB. (1) Phenol, (2) 2,5-dimethylphenol, (3) 4-chlorophenol, (4) 3,4-dimethylphenol, (5) 4-chloro-3-methylphenol, (6) 4-tert-butylphenol, (7) 2-tert-butyl-4-methylphenol, (8) metylparaben, (9) 4-pentylphenol, (10) ethylparaben, (11) isopropylparaben, (12) 2-phenylphenol, (13) 4-hexylphenol, (14) 4-tert-octylphenol, (15) propylparaben, (16) isobutylparaben, (17) 4-heptylphenol, (18) nonylphenol, (19) butylparaben, (20) 4-phenylphenol, (21) pentachlorophenol, (22) triclosan, (23) bisphenol F, (24) benzylparaben, (25) bisphenol A, (26) bisphenol B (27) bisphenol Z, and (28) bisphenol S.

**Figure 3 foods-12-02909-f003:**
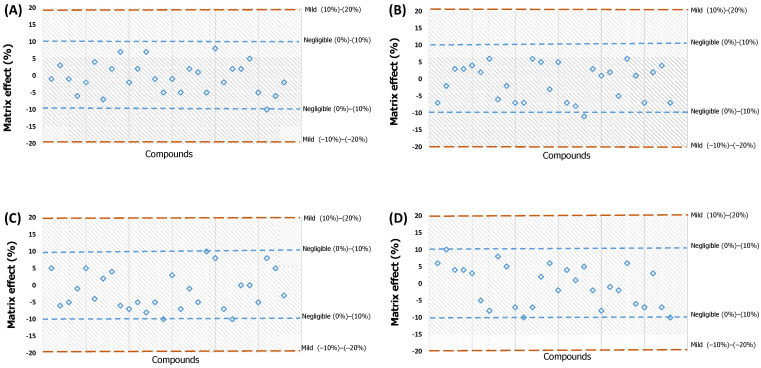
Matrix effect of parabens and phenolic compounds in milk (**A**), yoghurt (**B**), cheese (**C**), and butter (**D**) encountered with EMR-lipid and Oasis PRiME HLB in combination. The blue squares in the picture correspond to the *MFs* of the different analytes.

**Table 2 foods-12-02909-t002:** UHPLC-MS/MS parameters for the determination of the studied dairy product samples.

Compounds		t_R_ ^a^	Precursor (*m*/*z*)	Quantification Peak (*m*/*z*)	Collision Energy (V)	Confirmation Peak (*m*/*z*)	Collision Energy (V)	RF Lens (V)	LOD ^b^ (ng/kg)	Linearity (ng/kg)	r^2 c^
Parabens	Methylparaben	4.63	152.2	93.0	21.0	137.0	14.1	56.0	1.0	3.3–20,000	0.997
	Ethylparaben	6.57	166.2	93.0	22.6	138.0	14.0	50.0	1.0	3.4–20,000	0.996
	Isopropylparaben	9.48	179.3	93.0	21.6	137.0	14.3	49.6	1.0	3.3–20,000	0.994
	Butylparaben	9.66	197.0	129.1	11.7	178.1	12.5	51.8	1.1	3.6–20,000	0.996
	Propylparaben	10.11	180.2	93.0	23.3	92.0	23.8	53.8	1.1	3.6–20,000	0.995
	Isobutylparaben	15.29	193.2	92.0	24.8	137.1	15.7	51.3	10	34–20,000	0.997
	Benzylparaben	16.42	227.1	92.0	24.1	136.1	14.9	50.1	1.0	3.3–20,000	0.998
Phenols	Pentylphenol	0.74	165.0	96.9	10.3	78.9	26.7	37.4	1.1	3.5–20,000	0.997
	4-Chloro-3-methylphenol	0.97	141.0	35.0	19.1	105.0	16.7	62.4	10	34–20,000	0.994
	4-*tert*-Octylphenol	1.15	209.1	96.9	15.0	78.9	34.1	65.8	10	35–20,000	0.994
	Phenol	1.27	91.1	45.0	10.3	89.4	55.0	206.5	10	34–20,000	0.999
	4-*tert*-Butylphenol	1.48	147.1	61.9	11.4	46.0	45.2	35.7	1.1	3.5–20,000	0.995
	Nonylphenol	1.59	219.1	172.9	10.3	171.0	19.9	80.9	1.1	3.6–20,000	0.998
	2-*tert*-Butyl-4-methylphenol	1.69	147.1	61.9	13.9	89.0	12.8	30.0	1.0	3.3–20,000	0.997
	Bisphenol F	1.82	198.0	130.1	10.3	171.1	10.8	30.0	10	34–20,000	0.994
	3.4-Dimethylphenol	3.44	121.0	77.0	13.1	91.9	23.5	52.8	20	65–20,000	0.995
	2.5-Dimethylphenol	3.66	121.0	76.9	10.9	91.9	25.8	49.8	10	33–20,000	0.997
	4-Phenylphenol	4.19	169.0	19.2	32.6	69.0	54.1	38.7	1.0	3.3–20,000	0.995
	2-Phenylphenol	4.36	169.1	19.3	41.9	150.9	13.8	48.3	1.1	3.6–20,000	0.997
	Bisphenol S	4.51	249.1	108.0	28.0	92.0	35.3	109.1	1.0	3.4–20,000	0.997
	4-Chlorophenol	7.11	127.0	35.1	19.3	91.0	17.6	60.4	7.0	23–20,000	0.994
	4-Hexylphenol	16.74	177.1	148.0	16.0	149.1	16.8	158.8	10	34–20,000	0.995
	Bisphenol A	16.89	221.1	205.2	30.2	164.0	25.2	115.8	5.1	17–20,000	0.996
	Bisphenol B	19.08	241.1	225.0	10.2	223.0	10.2	68.6	5.0	16–20,000	0.994
	4-Heptylphenol	19.54	195.3	179.1	19.9	167.0	16.0	78.7	7.0	23–20,000	0.996
	Bisphenol Z	20.48	267.1	173.1	27.0	223.0	32.6	100.7	5.0	17–20,000	0.994
	Pentachlorophenol	25.8	264.8	35.0	55.0	97.0	27.4	125.4	10	34–20,000	0.994
	Triclosan	25.36	289.0	35.04	10.3	37.1	10.2	45.1	1.0	3.3–20,000	0.996
	Triphenylphosphate (IS)	27.54	337.3	96.9	32.7	79.9	46.5	122.9			

t_R_ ^a^: retention time; ^b^ LOD: limit of detection; ^c^ r^2^: correlation coefficient.

**Table 3 foods-12-02909-t003:** Results obtained from precision studies for the different types of dairy product samples.

Compounds		RSD (%) ^a^															
		Milk		Yogurt		Custard		Milkshake		Cream		Cheese		Butter		Margarine	
		Intraday	Interday	Intraday	Interday	Intraday	Interday	Intraday	Interday	Intraday	Interday	Intraday	Interday	Intraday	Interday	Intraday	Interday
**Parabens**	Methylparaben	5.11	8.22	5.31	10.21	5.05	10.16	5.06	9.03	2.55	9.75	4.26	9.26	4.01	7.12	2.36	7.42
	Ethylparaben	4.26	9.33	4.43	11.15	5.06	8.03	3.83	7.23	2.13	7.98	5.62	7.32	5.25	8.92	4.53	8.54
	Isopropylparaben	5.13	11.10	5.55	9.08	6.88	9.39	5.19	9.77	8.26	9.28	3.13	5.52	4.37	11.44	6.06	10.23
	Butylparaben	5.25	11.28	4.23	9.17	5.20	9.40	3.43	6.29	6.28	9.58	4.22	9.13	4.28	10.58	5.27	10.44
	Propylparaben	5.06	9.35	6.12	10.35	4.18	9.84	3.81	6.27	3.29	7.33	2.03	8.57	6.64	6.35	6.69	9.65
	Isobutylparaben	4.27	8.63	5.46	10.42	3.36	8.42	5.73	8.88	8.48	10.1	3.65	6.56	3.46	7.72	8.50	10.16
	Benzylparaben	3.19	9.46	5.88	9.44	4.32	8.03	6.52	9.06	5.55	9.08	4.42	9.04	4.56	6.96	4.04	6.34
**Phenols**	Pentylphenol	3.00	7.78	7.20	9.46	3.54	10.04	5.06	10.03	4.23	9.17	5.82	11.02	5.83	7.38	6.68	8.43
	4-Chloro-3-methylphenol	5.34	9.50	4.25	8.58	4.07	10.15	2.82	9.52	6.12	10.35	5.25	7.24	5.62	8.54	4.52	8.06
	4-*tert*-Octylphenol	4.16	8.44	2.13	7.40	6.48	9.35	8.07	10.44	5.46	10.42	6.52	8.33	5.75	8.94	3.06	8.03
	Phenol	3.42	7.65	5.52	9.06	4.27	10.13	5.08	8.32	7.26	11.03	3.21	7.75	3.06	6.25	2.82	9.82
	4-*tert*-Butylphenol	5.15	11.16	5.06	10.03	3.33	9.23	2.24	9.66	4.28	9.44	4.27	8.24	7.52	9.18	7.07	9.44
	Nonylphenol	7.06	10.23	4.82	9.82	4.66	7.55	3.55	9.63	5.35	9.38	4.17	9.12	5.54	10.07	4.01	7.10
	2-*tert*-Butylphenol	5.27	10.44	7.07	9.44	7.91	10.25	4.05	6.01	4.20	8.17	5.07	7.02	7.36	9.25	4.25	8.99
	Bisphenol F	4.69	9.65	4.08	8.22	4.12	8.02	3.06	8.03	2.93	7.89	4.56	6.45	3.12	6.32	5.37	11.04
	3,4-Dimehylphenol	7.50	10.16	4.24	9.66	6.32	9.33	4.75	9.95	3.05	8.53	3.04	6.34	4.23	6.64	6.28	10.38
	2,5-Dimethylphenol	3.56	7.76	3.55	9.13	7.84	10.07	9.41	7.91	5.27	10.26	4.68	9.43	4.45	6.87	7.64	6.33
	4-Phenylphenol	5.23	10.31	6.13	9.21	6.86	8.25	8.02	4.12	6.38	10.39	4.75	5.24	3.32	7.02	4.46	8.70
	2-Phenylphenol	4.45	9.22	7.26	11.03	4.12	7.32	9.06	6.32	5.20	11.40	3.37	7.25	3.55	9.74	5.56	7.95
	Bisphenol S	5.27	10.26	4.28	9.44	4.13	9.64	3.12	7.32	4.01	7.11	2.61	5.24	4.32	6.64	4.18	9.84
	4-Chlorophenol	6.38	10.39	5.29	10.77	6.32	9.33	7.13	9.64	3.25	7.99	4.32	8.03	2.58	7.54	3.36	8.42
	4-Hexylphenol	5.20	11.40	3.48	5.29	4.37	9.44	6.06	10.23	5.38	11.54	5.54	10.04	4.78	10.96	5.34	9.50
	Bisphenol A	4.18	8.84	3.86	6.27	3.28	10.58	9.82	4.66	5.98	9.38	5.07	10.15	7.63	9.68	7.26	11.03
	Bisphenol B	2.36	7.42	4.73	8.86	4.08	8.22	4.12	8.02	4.24	9.66	3.48	9.35	4.96	8.02	4.28	9.44
	4-Heptylphenol	4.53	8.54	4.32	7.22	3.36	6.76	3.05	9.03	3.55	9.13	5.27	10.13	5.88	9.03	2.36	7.42
	Bisphenol Z	4.66	9.65	6.23	8.41	4.18	8.84	3.86	6.27	4.12	7.32	3.33	9.23	4.24	8.85	5.20	11.40
	Pentachlorophenol	4.48	9.27	3.44	5.58	2.61	4.24	4.32	6.64	3.18	8.84	8.91	11.30	5.85	7.23	6.38	10.39
	Triclosan	5.00	10.16	5.36	9.63	4.13	9.64	4.05	9.05	4.05	9.51	3.12	8.22	5.16	10.41	5.20	11.40

^a^ RSD: relative standard deviation (*n* = 12) for 100 ng/kg (*p* < 0.05).

**Table 4 foods-12-02909-t004:** Results obtained in the recovery studies of the different types of dairy product samples.

Compounds			Recoveries (% ± SD, *n* = 3) ^a^						
		Milk	Yogurt	Cheese	Milkshake	Cream	Custard	Butter	Margarine
**Parabens**	Methylparaben	102 ± 8	100 ± 9	91 ± 8	95 ± 6	95 ± 7	95 ± 7	94 ± 6	91 ± 9
	Ethylparaben	103 ± 8	91 ± 6	95 ± 7	102 ± 7	102 ± 6	96 ± 9	98 ± 5	91 ± 9
	Isopropylparaben	95 ± 7	97 ± 5	95 ± 10	101 ± 6	101 ± 6	101 ± 9	104 ± 8	98 ± 10
	Butylparaben	100 ± 9	99 ± 4	99 ± 10	101 ± 10	104 ± 7	93 ± 6	100 ± 5	99 ± 5
	Propylparaben	104 ± 5	101 ± 6	95 ± 9	99 ± 9	99 ± 9	95 ± 6	104 ± 7	95 ± 9
	Isobutylparaben	96 ± 7	104 ± 7	105 ± 7	103 ± 9	99 ± 8	96 ± 11	99 ± 9	105 ± 7
	Benzylparaben	105 ± 6	105 ± 8	102 ± 8	91 ± 9	103 ± 6	95 ± 6	105 ± 5	100 ± 8
**Phenols**	Pentylphenol	103 ± 9	103 ± 5	101 ± 6	99 ± 9	98 ± 8	101 ± 6	100 ± 8	96 ± 11
	4-Chloro-3-methylphenol	96 ± 6	93 ± 6	100 ± 5	101 ± 10	100 ± 8	94 ± 7	103 ± 8	96 ± 11
	4-*tert*-Octylphenol	102 ± 7	92 ± 7	100 ± 5	96 ± 11	104 ± 8	95 ± 6	101 ± 8	106 ± 6
	Phenol	103 ± 9	97 ± 7	91 ± 8	100 ± 8	96 ± 6	95 ± 6	100 ± 7	103 ± 5
	4-*tert*-Butylphenol	101 ± 6	98 ± 8	97 ± 9	104 ± 8	94 ± 7	91 ± 9	103 ± 6	91 ± 8
	Nonylphenol	100 ± 8	91 ± 9	94 ± 7	102 ± 7	95 ± 6	96 ± 11	102 ± 7	104 ± 8
	2-*tert*-Butylphenol	104 ± 8	91 ± 9	95 ± 6	96 ± 6	100 ± 7	96 ± 6	103 ± 6	102 ± 7
	Bisphenol F	96 ± 11	99 ± 9	101 ± 6	96 ± 7	104 ± 7	96 ± 9	98 ± 8	97 ± 9
	3,4-Dimethylphenol	105 ± 7	99 ± 8	101 ± 10	101 ± 9	91 ± 8	98 ± 5	93 ± 7	94 ± 7
	2,5-Dimethylphenol	99 ± 8	103 ± 8	96 ± 9	98 ± 5	99 ± 5	104 ± 8	99 ± 9	98 ± 5
	4-Phenylphenol	102 ± 7	102 ± 7	101 ± 8	103 ± 9	102 ± 6	98 ± 5	96 ± 6	99 ± 9
	2-Phenylphenol	100 ± 7	96 ± 6	95 ± 6	103 ± 8	98 ± 5	99 ± 9	98 ± 8	96 ± 11
	Bisphenol S	98 ± 10	101 ± 9	106 ± 6	102 ± 7	95 ± 6	101 ± 8	94 ± 6	99 ± 9
	4-Chlorophenol	99 ± 5	100 ± 7	103 ± 5	98 ± 5	99 ± 9	95 ± 6	102 ± 9	94 ± 7
	4-Hexylphenol	102 ± 6	105 ± 8	96 ± 8	104 ± 8	96 ± 11	96 ± 6	102 ± 9	95 ± 6
	Bisphenol A	96 ± 9	101 ± 9	97 ± 7	98 ± 5	105 ± 7	98 ± 10	95 ± 7	100 ± 5
	Bisphenol B	101 ± 9	95 ± 7	97 ± 7	104 ± 8	98 ± 10	98 ± 5	105 ± 7	91 ± 8
	4-Heptylphenol	96 ± 9	102 ± 6	97 ± 6	97 ± 5	95 ± 10	104 ± 8	98 ± 5	103 ± 9
	Bisphenol Z	95 ± 6	95 ± 8	103 ± 6	99 ± 4	99 ± 10	97 ± 9	105 ± 10	96 ± 11
	Pentachlorophenol	99 ± 9	95 ± 8	99 ± 8	99 ± 8	103 ± 5	94 ± 7	97 ± 8	105 ± 7
	Triclosan	102 ± 5	97 ± 9	90 ± 8	102 ± 7	96 ± 8	104 ± 8	91 ± 9	104 ± 8

^a^ Percent recoveries (% ± SD, *n* = 3) of analytes spiked to milk and dairy product samples for 100 ng/kg.

**Table 5 foods-12-02909-t005:** Parabens and phenolic compound concentrations (mean ± standard deviation, ng/kg, *n* = 3) detected in different milk ^a^.

		Milk														
		Skimmed Cow’s Milk ^b^				Semi-Skimmed Cow’s Milk				Semi-Skimmed Goat’S Milk		Semi-Skimmed Sheep’s Milk	Whole Cow’s Milk			
**Detected** **compounds**	**Sample** **(Packaging Material)**	**1** **(MC) ^c^**	**2** **(MC)**	**3** **(MC)**	**4** **(HDPE)**	**1** **(MC)**	**2** **(MC)**	**3** **(HDPE)**	**4** **(PET)**	**1** **(MC)**	**2** **(MC)**	**1** **(MC)**	**1** **(MC)**	**2 ^e^** **(MC)**	**3** **(PET)**	**4** **(HDPE)**
	Nonylphenol	- ^d^	-	-	-	-	-	130 ± 10	34 ± 3	210 ± 20	-	53 ± 4	-	-	-	-
	4-*tert*-butylphenol	170 ± 10	-	-	-	81 ± 6	220 ± 20	110 ± 10	21 ± 1	220 ± 20	-	-	-	-	-	-
	Ethylparaben	80 ± 6	63 ± 3	47 ± 4	-	33 ± 2	-	-	-	470 ± 40	300 ± 30	210 ± 20	300 ± 20	-	-	-
	4-Chlorophenol	160 ± 10	110 ± 10	60 ± 5	-	50 ± 4	-	-	-	130 ± 10	-	-	-	-	-	-
	Propylparaben	-	-	-	-	-	10 ± 1	-	9 ± 0.8	33 ± 3	24 ± 2	23 ± 2	30 ± 2	-	-	-
	Benzyl-paraben	-	-	-	-	31 ± 3	24 ± 2	-	33 ± 1	98 ± 9	90 ± 6	86 ± 6	130 ± 10	-	-	-
	Bisphenol A	130 ± 10	33 ± 3	-	-	67 ± 5	-	-	-	64 ± 5	99 ± 8	170 ± 10	580 ± 50	260 ± 20	190 ± 10	290 ± 20
	Bisphenol Z	33 ± 3	44 ± 4	41 ± 4	25 ± 2	35 ± 3	34 ± 3	24 ± 2	36 ± 3	57 ± 5	47 ± 4	45 ± 4	42 ± 3	-	-	-
	Pentachlorophenol	-	-	-	-	-	40 ± 3	35 ± 3	55 ± 5	40 ± 2	-	76 ± 6	-	-	-	-

^a^ The compounds not listed were not detected. ^b^ (fat content,%)/(protein content,%): skimmed cow’s milk: (0.05–0.3%)/(3.2–3.4%); semi-skimmed cow’s milk (0.05–0.3%)/(3.2–3.4%); semi-skimmed goat´s milk (1.5–1.6%)/(3.3–3.4%); semi-skimmed sheep´s milk (1.6%)/(5.4%); whole cow´s milk (3.6%)/(3.0–3.2%); ^c^ HDPE: High density polyethylene; MC: multilayer-cardboard; PET: polyethylene terephthalate; PS: polystyrene. ^d^ -: not detected. ^e^ Fresh whole cow’s milk.

**Table 6 foods-12-02909-t006:** Parabens and phenolic compound concentrations (mean ± standard deviation, ng/kg, *n* = 3) detected in dairy products ^a^.

		Dairy Products															
		Milkshake			Yogurt				Cheese			Butter		Margarine		Cream	
					Sheep	Cow	Cow	Sheep									
**Detected** **compounds**	**Sample** **(packaging material)**	**1** **(PET)**	**2** **(HDPE)**	**3** **(MC)**	**1** **(Glass)**	**2** **(Glass)**	**3** **(PS)**	**4** **(Glass)**	**1** **(PS)**	**2** **(PS)**	**3** **(PS)**	**1** **(PS)**	**2** **(PS)**	**1** **(PS)**	**2** **(PS)**	**1** **(MC)**	**2** **(MC)**
	Nonylphenol	- ^d^	-	-	43 ± 3	-	-	-	-	-	-	-	98 ± 8	99 ± 7	49 ± 4	-	-
	Bisphenol A	360 ± 20	420 ± 40	370 ± 30	-	-	58 ± 5	-	-	150 ± 10	-	140 ± 10	50 ± 5	170 ± 10	160 ± 10	260 ± 20	450 ± 40

^a^ The compounds not listed were not detected. ^b^ (fat content,%)/(protein content,%): milkshake (0.8–0.9%)/(1.6–3%); yogurt (1.9–6.4%)/(3.1–5.5%); cheese (14–15%)/(10–16%); butter (80–82%)/(0.4–0.6%); margarine (60–70%)/(0.3%); cream (18–38%) (1.8–2.5%). ^c^ HDPE: High density polyethylene; MC: multilayer-cardboard; PET: polyethylene terephthalate; PS: polystyrene. ^d^ -: not detected.

## Data Availability

The data used to support the findings of this study can be made available by the corresponding author upon request.

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
