# Peer review of "Determination of Parabens and Phenolic Compounds in Dairy Products through the Use of a Two-Step Continuous SPE System Including an Enhanced Matrix Removal Sorbent in Combination with UHPLC−MS/MS"

_foods, 2023, doi:10.3390/foods12152909_

Round 1

Reviewer 1 Report

The study established a two-step continuous solid-phase extraction (SPE) and purification method for the determination of parabens and phenolic compounds in milk with UPLC-MS/MS. It is a interesting research. However, the manuscript is not publishable in the present form, since it was not well-designed and presented. It should be reject according to the points as follows:

General comment:

1. In the reviewer's version, some of the tables could not be fully displayed on one page, which prevented me from reading them completely.

2. The format of line number of this manuscript is disorganized, i can't express exactly which line of text. It is recommended to use consistent line numbers throughout the text, and to ensure that the line numbers can correspond to the text lines! It’s must be revised!

Introduction:

1. It is suggested that the method listed in Table 1 be compared with the research in this paper, and it is placed in the later content to show the advantages of the research method.

2. In last sentence of the six paragraph in introduction, the “EMRI-lipid” needs to be corrected to “EMR-lipid”, and the similar errors in the text should also be corrected.

Material and methods

1. How the two-step SPE system is activated, according to previous reports, the activation procedures of the two columns are not consistent. Please explain.

Result and discussion

1. In Figure 2, please use the serial number to indicate each conpounds.

2. In the preface, it is explained that there are EU regulatory limits for each substance, and why the LOQ is not compared with EU regulations or other methods. Mere listing of values does not indicate whether regulatory requirements are met.

3. In the section 3.3, it was said that the recovery rate was determined at three levels of spiked standards. Why is RSD only expressed at the lowest spiked level (100ng/kg) in the following text (Table 4)?

4. Intraday and interday precision are commonly used expressions.

Some words are misspelled and need to be corrected.

Author Response

Reply to the Comments of Reviewer 1

The authors would like to thanks the Reviewer for his/her apt comments and suggestion on their paper, which have helped them to improve it substantially. An itemized reply to his/her queries follows:

General Comment:

  1. The Reviewer is right. As suggested, the tables have been resubmitted. In addition, the publisher has been asked to maintain the visual quality of the tables in terms when formatting the manuscript.
  2. The line numeration in the manuscript depend on the automatic pdf generator by the publisher when submitted. However, the manuscript is also sent in word format with numbered lines.

Introduction:

  1. As suggested, the methods listed in Table 1 have been compared with the method proposed in Section 3.3. References to this Table are included in Sections 3.3 and 3.4.
  2. Sixth paragraph in Introduction: As suggested, “EMRI-lipid” has been corrected to “EMR-lipid”.

Material and methods

  1. According to the Reviewer’s suggestion, the activation procedures of the two columns are explained. For this purpose, the following sentences have been included in the last paragraph of Section 2.2. “The Oasis PRiME HLB sorbent column was conditioned by passing 1 mL of methanol, 1 mL of acetonitrile and 2 mL of Milli-Q water in this sequence. The EMR-lipid sorbent column was conditioned by passing 2 mL of Milli-Q water.”

Results and discussion

  • Figure 2. The serial number of each compound has been included in Figure 2. In addition, the figure caption includes what each serial number refers to.
  • According to the Reviewer’s suggestion, the EU regulatory limits have been compared with the LOQ of proposed method. So, the following sentence has been included in the Section 3.3: “Limits of quantification of proposed method (minimum value of the linear range) vary between 3.3 and 65 ng/kg. These values are lower than the EU regulatory limits of the legislated analytes as previously indicated [18]. Similarly, the LOQs of several of the methods included in Table 1 are lower than the EU regulatory limits [7,25,27]."
  • Section 3.3. As suggested, the results of RSDs and recoveries for 500 and 2000 ng/kg were included in the Table 2S (Supplementary material). The results of RSDs and recoveries for 100 ng/kg are included in Tables 3 and 4, as they were in the original version of the manuscript.
  • Table 3 and text. As suggested, “within-day” and “between-day” precision have been changed to “intraday” and “interday” precision, respectively.

Comments on the Quality of English Language.

The text has been reviewed by a native speaker.

Reviewer 2 Report

The manuscript “Determination of parabens and phenolic compounds in dairy products by use of a two-step continuous SPE system including an enhanced matrix removal sorbent in combination with UHPLC−MS/MS” presents a new method used for the determination of parabens and some phenols in dietary products. The topic fits the scope of the journal. The developed method seems to be appropriate for the intended use and sample preparation is reduced compared to other methods. However, the quality of the submitted manuscript is extremely low in terms of presented results because almost all tables are truncated. Therefore, it is impossible to judge the quality of the study including method validation.

There are many corrections to be done in the presented manuscript apart from corrections in tables (including visibility of their content, appropriate titles, and numbering).

Page 6 Explain what is “three three injection valves model”, and rewrite the text.

Page 6 PET is not polystyrene terephthalate. It is polyethylene terephthalate. The same relates to other pages of the manuscript (e.g. page 14 and the text below Table 5).

Page 7 Explain what is “two serially columns”, rewrite the text.

Figure 2 Why the line in Figure 2B is so high? It should be at 100%

Page 11 I cannot see precision results for the 3 mentioned concentration levels.

Page 11 There is no Table S1 in the submitted manuscript.

Page 11 I cannot see recovery results for the 3 mentioned concentration levels.

Page 11 Remove dates from publications mentioned in the text. You use numbers, so the dates are not needed. The same relates to other pages of the manuscript (e.g. page 14).

Table 4 Replace Milkshare with Milkshake.

Page 14 HDPE is not high-density polystyrene, it is high-density polyethylene.

Page 15 In Conclusion you wrote “The most detected phenolic compounds were BPA (33‒580 ng/kg) and BPZ (24‒57 ng/kg). In any case, these concentration levels do not exceed the legal limits set by the European Union.” Is there any European Union limit for bisphenol Z? The same text is also written in the abstract in a shorter incomprehensible version. It should be corrected in both Abstract and Conclusions.

Minor corrections of English are required.

Author Response

Reply to the Comments of Reviewer 2

First of all, the authors would like to thank the Reviewer for his/her comments on our paper. An itemized reply to his/her queries follows:

  • The Reviewer is right. As suggested, the tables have been resubmitted. In addition, the publisher has been asked to maintain the visual quality of the tables in terms when formatting the manuscript.
  • Page 6 and 7. According to the Reviewer’s suggestion, Figure S1 (Supplementary Material) is included to explain the operation of the three-injection valve and the sorbent columns. In addition, the operation is described. The following sentence is included in the manuscript to describe the operation of the continuous system "Figure S1 (Supplementary material) details the operation of the continuous system for the SPE.” (Last sentence of Section 2.4)
  • Page 6: As suggested, “polystyrene terephthalate” has been changed to “polyethylene terephthalate”.
  • Figure 2: As suggested, the line has been relocated in the correct place.
  • Page 11. As suggested, the results of RSDs and recoveries for 500 and 2000 ng/kg were included in the Table S2 (Supplementary material). The results of RSDs and recoveries for 100 ng/kg are included in Tables 3 and 4, as they were in the original version of the manuscript.
  • Page 11. Table S1 is included in Supplementary material. This clarification is included in the text.
  • Page 11. As suggested, the dates for publication in the test has been removed. The same relates to other pages of the manuscript.
  • Table 4. As suggested, “milkshare” has been changed to “milkshake”.
  • Page 14: As suggested, “high-density polystyrene” has been changed to “high-density polyethylene”.
  • Page 15. Conclusion. According to the Reviewer’s suggestion, the sentence has been modified as follows: “The most detected phenolic compounds were BPA (33‒580 ng/kg) and BPZ (24‒57 ng/kg). In the case of BPA, these concentration levels do not exceed the legal limit set by the European Union.”

Comments on the Quality of English Language.

  • The text has been reviewed by a native speaker

Reviewer 3 Report

The work of Colón and his/her colleagues is interesting about determination of parabens and phenolic compounds in dairy products by use of a two-step continuous SPE system including an enhanced matrix removal sorbent in combination with UHPLC−MS/MS. I would like to recommend it to major revision. There are some concerns as follows:

1.     In the section “3.3. Validation of the method”, the authors did not mention calibration curves and ranges of those analytes of interest, which are important to a new developed method, please add calibration curves and ranges.

2.     Some data of Table 1-5 is blocked because of layout problem, please provide complete data.

3.     The authors mention that the new method with only slight matrix effects, please provide chromatograms of solvent and sample to support your result.

4.     In the section “4. Conclusion”, the authors mention the new method can expedite sample treatment, but there is no relevant experiment in this paper, please provide your evidence.

5.     In the section “1. Introduction”,Please check the first abbreviation of gas chromatography.

6.     Please unify the nomenclature of breast milk in the table 1.

OK

Author Response

Reply to the Comments of Reviewer 3

The authors would like to thanks the Reviewer for his/her apt comments and suggestion on their paper, which have helped them to improve it substantially. An itemized reply to his/her queries follows:

General Comment:

  1. The following sentence was included in the original manuscript: "The method provided a linear response over the concentration range 3.3-20 000 ng/kg and correlation coefficients (r2) exceeded 0.994 for all analytes". In addition, the linear range data and the correlation coefficients of the calibration curves for each analyte are included in Table 2.
  2. The Reviewer is right. As suggested, the tables have been resubmitted. In addition, the publisher has been asked to maintain the visual quality of the tables in terms when formatting the manuscript.
  3. As suggested, the Figure S2 (Supplementary material) has been included where 2 chromatograms are shown. In one chromatogram, the analyte is spiked in the solvent, and in the other the analyte is spiked in a milk sample. Figure S2 is also referred to in the text.
  4. As suggested, the following sentence has been added: “Thus, the proposed method speeds up the sample treatment by reducing the time of this step to 15 min compared to other methods that need more than 40 min [4,24,25].”
  5. Introduction: As suggested, “CG” has been changed to “GC”
  6. The nomenclature of breast milk in the Table 1 has been unified.

Round 2

Reviewer 2 Report

The manuscript “Determination of parabens and phenolic compounds in dairy products by use of a two-step continuous SPE system including an enhanced matrix removal sorbent in combination with UHPLC−MS/MS” has been corrected. However, still some tables are truncated – I cannot see LOD and r2 in Table 2, the results for margarine in Table 3, and the results for both butter and margarine in Table 4.

I know that English has been corrected by a native speaker, but:

Page 6. I cannot understand the sentence “Previous analysis the individual analytes were infused and characterized, and multiple reaction monitoring (MRM) transitions optimized for each compound.” I am missing any sense in this sentence. I also wonder what earlier analysis you have in mind. A citation referring to the previous analysis would also be helpful.

Page 6. The extraction system was composed of a few parts, so you should replace “composed by” with “composed of”.

Page 6. In the last line o point 2.2 remove the sign “.

no comments

Author Response

First of all, the authors would like to thank the Reviewer for his/her comments on our paper. An itemized reply to his/her queries follows:

  • The Reviewer is right. As suggested, the Tables 2, 3 and 4 have been resubmitted. In addition, the publisher has been asked to maintain the visual quality of the tables when formatting the manuscript.
  • Page 6. According to the Reviewer’s suggestion, the sentence has been modified as following: “The individual analytes were infused and characterized in the mass-spectrometer triple quadrupole equipment in multiple reaction monitoring mode (MRM) (Table 2).” We mean that prior doing anything in the UHPLC-MS/MS equipment the analytes were characterized. There is not any previous analysis.
  • Page 6: As suggested, “composed by” has been changed to “composed of”."

Reviewer 3 Report

All my questions have been responded appropriately.

Author Response

The authors would like to thanks the Reviewer for his/her apt comments and suggestion on their paper.